# Knowledge, attitude and risky practices on schistosomiasis in Ethiopia: A scoping review

**Getaneh Alemu**[ID][1]*, **Endalkachew Nibret**[2,3], **Abaineh Munshea**[2,3], **Melaku Anegagrie**[4], **Arancha Amor**[ID][4]

**1** Department of Medical Laboratory Science, Bahir Dar University, Bahir Dar, Ethiopia, **2** Biology Department, Science College, Bahir Dar University, Bahir Dar, Ethiopia, **3** Health Biotechnology Division, Institute of Biotechnology (IoB), Bahir Dar University, Bahir Dar, Ethiopia, **4** Mundo Sano Foundation and Institute of Health Carlos III, Madrid, Spain

* getanehmlt@gmail.com

## Abstract

### Background

Despite many years of intervention measures, schistosomiasis (SCH) remains a public health problem in Ethiopia. Health education and promotion enable community involvement and active participation in SCH control and prevention. Therefore, it is considered as one of the key strategies to prevent and control SCH in Ethiopia. However, comprehensive data on the knowledge, attitude and practice (KAP) of vulnerable populations towards the disease are lacking. Therefore, we reviewed the existing KAP studies in Ethiopia.

### Methods

Studies conducted in Ethiopia and published between 2006 and 2023 were searched and reviewed from January to April 2024. Electronic literature searches were made in PubMed, Hinari, African Journal Online and Google Scholar using the keywords "Schistosomiasis, *Schistosoma*, *Schistosoma mansoni*, *Schistosoma haematobium*, Knowledge, Attitude, Practice, Perception, Belief, Ethiopia" by combining them with Boolean operators (AND, OR). The review was conducted according to the Arksey and O'Malley Framework for scoping reviews, and studies were selected based on the PRISMA guidelines. Thematic analysis was applied to summarize, synthesize and report results.

### Results

Ten studies that recruited 4,763 participants were included in the present review. Knowledge gaps on the source of *Schistosoma* infection, transmission, morbidity, treatment, and prevention in Ethiopia were identified. Studies have found large differences in attitudes toward SCH in terms of the population at risk, the severity of

**Data availability statement:** All relevant data are within the manuscript and its Supporting Information files.

**Funding:** The author(s) received no specific funding for this work.

**Competing interests:** The authors have declared that no competing interests exist.

**Abbreviations:** FMoH, Federal Ministry of Health; KAP, Knowledge Attitude and Practice; MDA, Mass Drug Administration; NTDs, Neglected tropical diseases; PSAC, Preschool-aged Children; SAC, School-aged Children; SBCC, Social Behavioral Change Communication; SCH, Schistosomiasis; SSA, sub-Saharan Africa; STHs, Soil-transmitted Helminths; WASH, Water Sanitation and Hygiene; WHO, World Health Organization.

the disease, and beliefs in the availability and success of its treatment and prevention. Furthermore, in most studies included in this review, the majority of participants had negative attitudes towards SCH. The majority of participants also engaged in risky water-related practices, which facilitated the ongoing transmission of SCH. KAP levels among community members, school-aged children, and mothers/caregivers of preschool-aged children showed no significant differences.

## Conclusions

The results of this systematic review showed that the KAP level is inadequate despite health education platforms that have been established and implemented for many years. Therefore, we recommend strengthening the implementation of health education and continuous monitoring of SCH prevention and control activities.

---

## Background

Schistosomiasis (SCH) is one of the 20 neglected tropical diseases (NTDs), and it is caused by trematodes of the genus *Schistosoma* [1]. The disease is endemic in 70 developing countries, and more than 200 million people are infected worldwide. *Schistosoma mansoni* and *S. haematobium* are the most widespread species both worldwide and in Africa. Sub-Saharan Africa (SSA) has the highest burden of SCH cases (90% of the infections), and up to 20 million people suffer from severe chronic health consequences of the disease. The disability-adjusted life years lost due to SCH are estimated to be 3.3 million [2].

Ethiopia is one of the countries in SSA with the highest burden of NTDs, including SCH [3]. A total of 966 districts were mapped between 2015 and 2020. Of these, 480 were endemic for SCH, and approximately 53.3 million people were at risk of infection. The intestinal (caused by *S. mansoni)* and urogenital (caused by *S. haematobium)* forms of SCH are common in Ethiopia [4]. Aware of this fact, the Federal Ministry of Health (FMoH) launched the first NTDs control program and implemented it between 2013 and 2015 [5]. During this period, 81.3% (10 million out of 12.3 million) of affected school-aged children (SAC) were dewormed with praziquantel. However, major challenges to sustaining interventions include lack of adequate attention to social behavioral change communication (SBCC), poor supply of clean water, and low level of sanitation and hygiene practices. The second National NTD Strategic Plan, which aimed to eliminate SCH to the point where it no longer poses a public health problem by 2020, was implemented from 2016 to 2020. Although 27 million SAC were treated during this period, SCH still represents a public health problem. Currently, Ethiopia is implementing the third National NTD Strategic Plan (2021–2025), which aims to eliminate *Schistosoma* transmission in Ethiopia by 2025. Mass drug administration (MDA), facility-based case diagnosis and treatment, snail control in hotspots, SBCC, and access to safe water supply, basic sanitation, and hygiene (WASH) are the five identified strategies to achieve this goal [4]. In addition to the national NTD strategic plans, the Health Extension

Program, which began operations in Ethiopia in 2004–2005 and is currently being implemented, aimed to engage the community in disease prevention through the transfer of health knowledge and skills to households [6]. However, preventable diseases such as SCH, soil-transmitted helminthes (STHs), and malaria continued to pose major public health problems [4,7].

Several studies have shown that reinfection after MDA is the main reason for persistent *Schistosoma* transmission in many endemic areas [4,8,9]. Therefore, MDA should be implemented in integration with SBCC, WASH, and vector control programs. Although SAC, women, and other people whose occupations involve contact with natural waters are at higher risk of infection [10,11], the intervention programs mainly target SAC. Schistosomiasis prevention targeted at all vulnerable populations would have enabled successful control of the disease [12].

Successful implementation of SCH control and prevention strategies requires commitment and active community participation. Therefore, health education should ideally occur before programs that require full community involvement, as this will enable the community to make informed decisions about their participation [13]. Cognizant of this fact, the World Health Organization (WHO) proposed as early as 1985 that health education and health promotion should be the focus of SCH prevention work [14]. However, a systematic review of articles published in SSA between 2006 and 2016 found that the KAP level greatly varied between studies [15]. Although some studies were conducted in Ethiopia, they were not included in the previous review [16–18], which may be due to strict eligibility criteria and review period.

Health education was considered as one of the key strategies to control and eliminate SCH in previous Ethiopian national NTD strategic plans. However, changes in the population's KAP towards SCH have not been sufficiently taken into account. Even the existing information is fragmented, and there is no updated comprehensive national data in this regard. Therefore, the aim of this review was to map the existing evidence about the KAP level in relation to SCH among different population groups in Ethiopia. This scoping review will help stakeholders assess the impact of the ongoing health education programs.

## Methods

### Search strategy

We conducted a search for articles published between January 2006 and December 2023. We excluded studies published before 2006 just to assess KAP level progress towards SCH in Ethiopia after the launch of the Health Extension Program (2004–2005). Two reviewers (GA and EN) independently searched literature in the databases PubMed, HINARI, African Journals Online, and Google Scholar, using the keywords "Schistosomiasis OR *Schistosoma* OR *Schistosoma mansoni* OR *Schistosoma haematobium* AND Knowledge OR Attitude OR Practice OR Awareness OR Perception OR Belief AND Ethiopia." The search was carried out by starting with broad terms, then narrowing them down step by step and combining keywords with Boolean operators (AND, OR). Additional searches were made based on the references listed in the identified publications. We conducted and reported this review in accordance with both the Arksey and O'Malley Framework for scoping reviews [19] and the Preferred Reporting Items for Systematic Reviews and Meta-Analyses (PRISMA) guideline (Fig 1) and checklist (S1 File). The search was peer-reviewed following the PRESS standard.

### Study selection

Study selection was performed through a three-stage review process: database search, review of abstracts by two authors, and eligibility review, as shown in the PRISMA flowchart and S2 File. The present study included studies conducted in Ethiopia between 2006 and 2023 that focused on a population's KAP to intestinal or urogenital schistosomiasis and were published in English. Studies published before 2006 were excluded. Selected articles were carefully read to find the best available data to achieve the objective of this review.

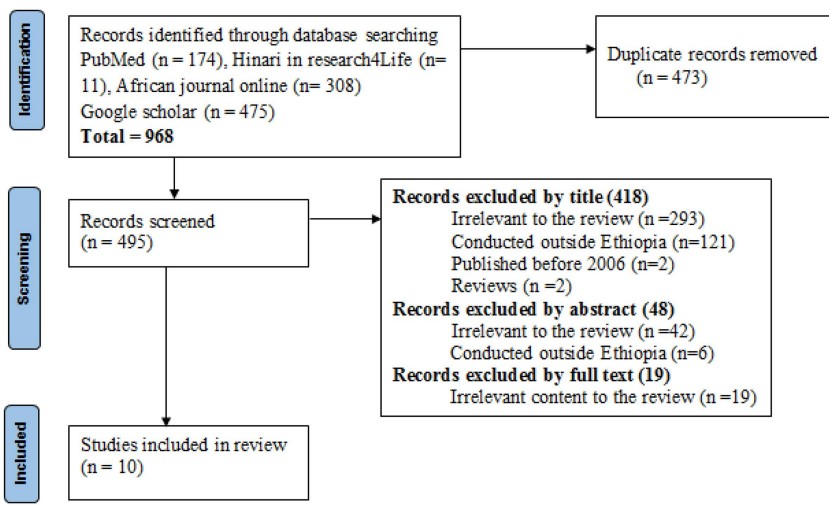

**Fig 1. PRISMA flow chart.**

## Quality assessment

A quality assessment of the included studies was carried out using a 'critical appraisal tool for the assessment of cross-sectional studies', which was developed by Downes *et al.* [16]. The studies were assessed based on the following indicators: (1) clear definition of objectives; (2) study design consistent with the stated objectives; (3) sample size justified; (4) target population clearly defined (appropriate population base/unbiased sampling); (5) correctly measured risk factor and outcome variables using instruments that have been previously tested, piloted, or published; (6) methods (including statistical methods) sufficiently described to permit replication; (7) results for analysis described in the methods presented; (8) author discussions and conclusions supported by results; (9) limitations of the study discussed; (10) ethical approval or consent of participants. Responses for each indicator were scored 0 for "no" and 1 for "yes". The quality of papers was classified as low, medium, and high if the total scores were between 1–4, 5–7 and 8–10, respectively (S3 File). Accordingly, six and four studies were of medium and high quality, respectively.

## Data extraction

We extracted data about the first author, year of publication, study aim, study design, study population, study location, and key study results in a table (S4 File). Thematic analysis was used for summarizing, synthesizing and reporting results.

## Results

A total of 968 articles were retrieved from all searched databases and websites. 473 articles were removed because they were duplicates. An additional 418, 48 and 19 articles were removed after reading the titles, abstracts and full texts, respectively, for the reasons mentioned in Fig 1. Therefore, 10 articles were included in the present review, all of which were cross-sectional studies. After a detailed reading of each included study, four themes were identified: 1) Sociodemographic characteristics; 2) Knowledge about SCH; 3) Attitude towards SCH; 4) Practices related to the prevention and control of SCH. Seven studies were community-based cross-sectional studies [17,20–25]; one study was conducted among SAC [18], while the remaining two studies recruited women with preschool-aged children (PSAC) [26,27]. Three studies were conducted before the launch of the national school-based MDA in 2015 [17,20,26], while data for the remaining seven studies were collected after the launch of the MDA [18,21–25,27]. The studies included in the present review

recruited 4,763 participants (2,118 male and 2,645 female) aged between 5 and 86 years old [17,18,20–22,24–27], although one study did not report the maximum age limit [23]. Data on KAP related to SCH were collected using a structured questionnaire, except for one study in which focus group discussion was conducted [23]. The ten studies reviewed are summarized in Table 1.

## Knowledge about schistosomiasis

Six out of the ten studies reported the proportion of respondents who had ever heard of the disease SCH/bilharziasis. In four articles, the majority of respondents – 87.2% [23], 80.8% [27], 71.5% [17] and 51.6% [18] – had never heard of SCH. Two studies reported high levels of awareness among participants: 97% [21], and 69% [25] had heard of the disease. Three studies reported the sources of information, with healthcare facilities/health professionals, family/friends and schools being most common [21,23,27].

In most studies, there was little knowledge about the source and modes of infection, and the majority of respondents answered "I don't know" when asked about both issues [17,20–22,26,27]. Related to the source of SCH infection, three studies reported good knowledge of SCH infection as the majority of the respondents mentioned contaminated water [17,23,27], while three others reported poor knowledge [20,25,26]. About the modes of SCH transmission, adequate knowledge was reported in only one study, in which 88.6% (31/35) of participants responded that they were exposed to infection while swimming or bathing in contaminated water [23].

Of the participants who were aware of SCH, 28.9% [21], 50.4% [17], 60.4% [23], 79% [22], and 94% [27] did not know at least one sign or symptom of SCH. In addition, most participants who knew the signs and symptoms were unable to comprehensively list the most common signs and symptoms. Respondents were asked, "Is schistosomiasis treatable?" in four studies. The proportion of participants who correctly answered was 21% [22], 29% [23], 78% [17], and 97.3% [21] among participants who had ever heard of SCH. In three studies, the majority of respondents – 66.7% [17], 79.6% [20], and 72.1% [21] – knew that SCH is preventable, while the other three studies reported poor knowledge, as 79% [22], 87.2% [23], and 94% [27] of the respondents did not know whether SCH can be prevented or not.

## Attitude towards schistosomiasis

A significant number of respondents believed that drinking dirty water, consuming contaminated food, air, mosquitoes, and flies are sources of *Schistosoma* infection [17,18,20,21,26]. One study reported that participants believed that eating contaminated food (23.7%), dirty hands (16.2%), and playing with soil (20%) predispose them to *Schistosoma* infection [18]. In one of the articles, 85% of respondents believed that intestinal SCH could not be transmitted by swimming or bathing in the river, crossing the river with bare foot, washing clothes in the river, or fishing in the river [27]. In another study, only 22.9% (11/48) of respondents agreed that there was a risk of *Schistosoma* infection from their contact with river or stream water. In the same study, the majority, 93.7% responded that they were not sure whether SCH was a serious illness or not [23]. However, in two studies, 96% [21] and 94% [24] of the participants understood SCH as a serious illness. In the context of treatment and preventive chemotherapy, in one study, 67% of the participants agreed on the importance of MDA, but only 2.8% (3/109) agreed that mass therapy was effective for SCH control [23]; in another study, approximately 99.7% agreed on the importance of medication during SCH [21]. Finally, the majority of 94% [21] and 66.7% [17] of the participants who were aware agreed that SCH is preventable, while 89.6% [23] and 79% [22] in other studies did not agree with this fact.

## Risky practices for *Schistosoma* infection

Many of the participants had frequent contact with freshwater while swimming or bathing, washing clothes and utensils, crossing rivers barefoot or fetching freshwater for household use [17,18,20,21,23,25]. In two studies, 98% [20] and 56.5%

**Table 1.** Summary of studies in this review.

| Author/year | Study objectives | Type of study | Population/ study location | Main findings |
|---|---|---|---|---|
| Mengstu et al., 2009 [17] | To assess the level of awareness about intestinal Schistosomiasis in communities living in intestinal schisto-somiasis endemic areas of Ethiopia | Cross-sectional study using open and closed-ended questionnaire | 417 community members (176 from Dudicha (aged15–70) and 241 from Shesha Kekel (aged 15–80) peasant associations) | • 123 (29.5%) heard about bilharzia<br>• 96 (78%) suggested water as source of infection<br>• Other sources of infection mentioned: poor sanitation (2, 1.6%), contagious (2, 1.6%), air (1, 0.8%), mosquito bite (1, 0.8%), do not know (21, 17.1%)<br>• 62 (50.4%) mentioned water contact as way of transmission<br>• 49 (39.8%) understood that bilharzia can be transmitted by both drinking con-taminated water and during bathing or swimming<br>• 53 (43.1%) knew abdominal pain or bloody diarrhea is a common symptom<br>• 96 (78%) knew that bilharzia has treatment<br>• 27 (22%) had no information whether bilharzia has treatment<br>• 108 (87.8%) knew that the disease affects children and adults as well as males and females<br>• 3 individuals from Shesha Kekel mentioned only children would catch bilharzia<br>• 12 (9.8%) individuals were not sure which age groups are affected<br>• 7 (5.7%) have knowledge about the intermediate host<br>• 2 (0.83%) from Shesha Kekel mentioned bilharzia as a common disease in the area<br>• 82(66.7%) knew that it is possible to prevent bilharzia by giving treatment, using clean water for drinking and washing and reducing water contact<br>• 94/96 (97.9%) practice frequent water contact |
| Nyanteki et al., 2010 [26] | To assess the mothers' (having <5 children) aware-ness about the cause, effect, mode of transmission, and preventive methods | Cross-sectional study using open-ended questionnaire | 130 mothers of PSAC from Shesha kekel | Mothers' response when asked 'How do children get bilharzia?'<br>Drinking dirty/river water (19, 16.4%)<br>Washing in river water (8, 6.9%)<br>Bad air, contaminated food or poor sanitation (13, 11.2%)<br>Do not know (82, 70.7%) |
| Nyanteki et al., 2014 [20] | To assess the knowledge of Abay Deneba village community | Community based cross-sectional study using structured and open ended questionnaire | 345 Household members | • Knowledge about the cause of SCH: contaminated water (64, 18.6%), contami-nated food (4, 1.2%), don't know (63, 18.3%), bath in river (5, 1.5%)<br>• Knowledge about common symptoms: abdominal discomfort (13, 3.8%), back pain, headache (3, 0.9%)<br>• Knowledge about preventive methods: avoid using contaminated water (49, 14.2%), don't know (70, 20.4%)<br>• 72.2% said their children had frequent contact with water bodies<br>• 60% said their children bathed in different sources of water<br>• 95.4% use water from Lake Ziway for drinking<br>• 98% defecate in the open field<br>• 89.3% did not receive health education |
| Alemu et al., 2016 [27] | To assess the prevalence of *S.mansoni*, STH infections and associated risk factors among preschool-aged children in Denbia district, North West Ethiopia. | Community based cross-sectional study using questionnaire | 401Mothers of PSAC | • 324 (80.8%) had never heard about intestinal SCH<br>• Major source of information was health professionals<br>• 85% responded that intestinal SCH cannot be transmitted by swimming or bathing in the river, crossing river with bare foot, washing clothes in river and fishing in river<br>• 380 (94.8%) of mothers did not know ways of prevention of intestinal SCH<br>• 377 (94%) did not know the symptoms of the disease |

*(Continued)*

| Author/year | Study objectives | Type of study | Population/ study location | Main findings |
|---|---|---|---|---|
| Gebreyohanns *et al.*, 2018 [21] | To determine the prevalence of intestinal parasites and KAPs among individuals who have river water contact with special emphasis on *S. mansoni* in the town of Addiremets, Ethiopia. | Community based cross-sectional study using questionnaire | 301 HH members >15 years old | • 292 (97%) heard about SCH<br>• Source of information: health facility (123, 43.4%), friends (82, 29%), school (67, 23.7%), radio (11, 3.9%)<br>• When asked about ways of transmission: swimming in infested river water (132, 43.9%), drinking dirty water (171, 56.8%), playing in infested water (43, 14.3%), snail (18, 6%), contaminated food (35, 11.6%), don't know (36, 12%)<br>• When asked about the sign and symptoms: fever (61, 20.3%), headache (8, 2.7%), weakness (81, 26.9%), dry cough (1, 0.3%), abdominal pain (155, 51.5%), diarrhea (102, 33.9%), blood in stool (79, 26.2%), do not know (87, 28.9%)<br>• 293 (97.3%) knew that SCH is treatable<br>• 283 (94%) knew that SCH is preventable<br>• When asked about prevention methods: treatment (54, 19.1%), Avoid bathing or swimming in stagnant water (128, 45.2%), use of toilet (64, 22.6%), provision of safe water (198, 70%), avoid open defecation (13, 4.6%), personal hygiene (13, 4.6%), do not know (1, 0.4%).<br>• 289 (96%) believe SCH is a serious disease<br>• 300 (99.7%) believe medication against SCH is important<br>• 258 (85.7%) think swimming/bathing in river water can cause SCH<br>• 296 (98.3%) think SCH is treatable<br>• 279 (92.7%) wash clothes in river<br>• 265 (88%) swim/bath in river<br>• 170 (56.5%) defecate around rivers<br>• 29 (9.6%) fetch river water for drinking/cooking |
| Mohammed *et al.*, 2018 [22] | To assess community awareness of *S. mansoni* in Haradenaba and Dertoramis kebeles in the Bedeno district, eastern Ethiopia. | Community based cross-sectional study using structured questionnaire | 572 participants of age ≥ 18 | • 466 (81.4%) did not know at risk population groups for S. mansoni<br>• 452 (79%) did not know the possible sources of infection<br>• 494 (86.4%) did not know the possible modes of transmission<br>• 452 (79%) did not know sign and symptoms<br>• 452 (79%) did not know if SCH is treatable<br>• 452 (79%) did not know if SCH is preventable |
| Assefa *et al.*, 2021 [23] | To investigate the levels of schistosomiasis related to KAP of schistosomiasis endemic community in the remote lowlands of the Abbey and Didessa Valleys in Benishangul Gumuz Region | A cross-sectional multilevel triangulation-mixed methods | 376 participants | • 328 (87.2%) never heard of SCH. 48 (12.8%) heard about SCH<br>• Main sources of information were family or friends (45.8%) and schools (36.7%), health facilities (20.4%).<br>• When asked about causes of SCH said that the cause for SCH is worm biting (62.5%, 30/48), drinking dirty water (8.3%, 4/48), eating food long time after cooked (6.3%, 3/48), did not know (22.9%, 11/48)<br>• 73% (35/48) knew the ways of SCH transmissions<br>• 88.6% (31/35) said while swimming/bathing in water bodies, 62.9% (22/35) said contact with worm and 22.9% (8/35) said drinking contaminated water.<br>• 39.6% (19/48) knew the symptoms of schistosomiasis and all of them mentioned itching as the most common symptom, blood in stool (5.3%), abdominal pain (42.1%), swollen abdomen (5.3%)<br>• 81.2% did not know ways of prevention. 9 (18.8%) knew it is preventable<br>• Prevention methods: avoiding contact with water bodies (6/9), MDA (7/9)<br>• 22.9% (11/48) agree that they were at the risk of Schistosoma spp. infection during contact with river or stream water bodies<br>• 2.1% (1/48) believed SCH is a serious disease, 93.7% did not know if it is serious<br>• 10.4% agree SCH is preventable<br>• 0% believe human excreta is not a source of infection<br>• Among 109 participants who were aware of MDA, 73 (67%) agree with its importance, but only 3 (2.8%) agree that MDA is effective<br>• Majority cross water bodies on the way to school or work<br>• 38.6% had gone to river/stream at least once per day to swim/bath or fetch water<br>• 87.5% said they have latrine but rivers and stream sides were full of human excreta |

*(Continued)*

**Table 1.** (Continued)

| Author/year | Study objectives | Type of study | Population/ study location | Main findings |
|---|---|---|---|---|
| Meleko *et al.*, 2023 [18] | To assess the prevalence of SCH and STH and examine the association between these diseases KAP of schoolchildren in the newly established Gidi Bench district, of Bench Sheko Zone, Southwest Ethiopia | School based cross sectional study | 611 SAC | • 302 (49.4%) heard about intestinal parasitosis<br>• When asked about how SCH is transmitted: eating contaminated food (23.7%, 145), dirty hands (16.2%, 99), Swimming or bathing in infested water (16.7%, 102), playing with soil (20%, 122)<br>• 40.8%% (249) swim/bath/play in river<br>• 59.4% (363) wash clothes or utensils in open water sources |
| Sule *et al.*, 2022 [24] | To evaluate theatre-based behavior change approach for influencing community uptake of SCH control measures | Community based follow up study | 369 participants | • 75% knew SCH affect both children and adults<br>• 50% and 44% knew SCH is very dangerous and dangerous, respectively<br>• 95% believe health facility is the right place to get treatment |
| Micho *et al.*, 2020 [25] | To assess the magnitude of intestinal parasites among Zay people residing in three islands of Lake Ziway in Ethiopia. | Community based Cross-sectional study | 444 people in Ziway | • 31% never heard about bilharziasis.<br>• 36% had no information about waterborne and related diseases<br>• 85.6% use lake water for cleaning, drinking or both |

[21] of the participants who had the awareness frequently defecated in the open field and near rivers. In one study, 87.5% of the respondents reported having a latrine, but rivers and stream sides in the study area were full of human waste during field observations [23]. Furthermore, in one study, 89.3% reported that they had never received health education on SCH prevention [20].

## Discussion

This systematic review assessed knowledge, attitudes, and practices towards SCH among community members, SAC, and mothers/caregivers of PSAC in Ethiopia. Although the SCH control and prevention programs are centrally designed and intended to be implemented along with MDA across the country [4], this review has shown differences in the KAP levels depending on geographical settings and study populations. In addition to inadequate health education and promotion, sociocultural factors could also pose barriers to changes in attitude and practices regarding SCH control and prevention [15].

According to this review, there is a lack of comprehensive knowledge about sources of infection, routes and modes of transmission, morbidity, treatment, and prevention of SCH in Ethiopia. This is unfortunate, as health promotion for infectious diseases of public health importance has been carried out under the Health Extension Program since 2004–2005 [6]. Furthermore, SCH was identified as one of the five chemotherapy-preventable NTDs in Ethiopia. Chemotherapy was accompanied by health education both at school and in the community [4]. The lower level of knowledge could be due to the inadequacy of health education sessions. For example, we observed that school- and community-based education sessions were conducted only at the time of deworming, which is an inadequate method to gain good knowledge and change community behavior. If properly implemented, health education has a high input-output ratio and is a cost-effective

prevention and control method. Comprehensive knowledge contributes to a positive attitude, which in turn contributes to behavioral changes such as reducing risky activities [28]. However, behavioral change will occur only if health education is accompanied by safe water supplies and basic sanitation [29,30]. This issue was identified in this review, where participation in risky activities (frequent water contact) was high regardless of the participants' knowledge level [17,18,20–27].

Misconceptions about SCH transmission and prevention, as well as low levels of knowledge, can lead to poor prevention practices [31,32]. In Ethiopia, MDA and health education programs for SCH and STHs are implemented together. We understood that there is confusion regarding the transmission and prevention of SCH and STHs [17,18,20,21,26]. For example, in one study, participants believed that consuming contaminated food (23.7%), dirty hands (16.2%), and playing with soil (20%) predispose them for the infection [18]; although these actually predispose them to STH infection rather than SCH infection. Therefore, dissemination of the correct information specific to each disease should be particularly emphasized as part of health education programs that include the control of more than one parasite (since it concerns STH together with SCH) and will be a key to enabling people to apply appropriate preventive measures [33]. Misunderstandings due to false or mixed messages can lead to false perceptions/beliefs in the community [34], which in turn impacts their health-seeking behaviors and participation in infection prevention [35].

Differences in health education implementation at different transmission settings and study populations could lead to differences in behavioral changes in Ethiopia. For example, in two studies, the majority, 94% [21] and 66.7% [17], of aware participants agreed that SCH is preventable, whereas 89.6% [23] and 79% [22] in other studies did not agree with this. There were also differences in attitudes towards at-risk populations, severity, treatment, and prevention across different studies [17,18,20–27].

The real purpose of health education is to impart knowledge about SCH and convince the community of their commitment to disease prevention and control. However, in the reviewed studies, the majority of participants were involved in risky activities regardless of their knowledge and attitudes towards SCH [17,18,20–27]. This could be due to the fact that health education was not complemented by basic WASH facilities and health services [15].

## Strengths and limitations of the review

We have for the first time presented a comprehensive overview of the KAP status regarding SCH in Ethiopia. However, the present review has the following limitations: The proportion of participants with good knowledge, positive attitudes, and good prevention practices would have been quantified based on the average scores of the questionnaire items using Bloom's modified cut-off points [36]; we were unable to do this because of great variations of the items used to assess KAP and, hence, a lack of standard definition of knowledge, attitude and practice. Therefore, the interpretation of the KAP results was subjective. This also made it difficult to compare KAP values between study results. There was wide variation in the depth of data presented in the reviewed studies such that studies that presented more data were overrepresented in the review. Due to the limited number of studies, it was difficult to draw definitive conclusions about the KAP levels in Ethiopia.

## Conclusions

The results of the reviewed studies showed that people's KAP levels were inadequate despite both community- and school-based health education platforms having been established and implemented for more than a decade. This brings to mind the future strengthening of the health education program and the monitoring of the practical implementation and impact assessment of the existing programs. Furthermore, the joint implementation of two programs together (namely, STH and SCH) should be precise enough to avoid confusion and mixed information. Preparation of a standard context-specific KAP assessment tool could be an immediate responsibility of the FMoH NTD Prevention and Control Program. In addition, further studies at both school and community levels are recommended to examine the KAP levels and associated factors in different endemic areas of all vulnerable populations.

## Supporting information

**S1 File. PRISMA 2020 checklist.**
(DOCX)

**S2 File. List of screened articles and their eligibility result.**
(XLSX)

**S3 File. Quality assesment results for reviewed studies.**
(DOCX)

**S4 File. Extracted data table.**
(DOCX)

## Author contributions

**Conceptualization:** Getaneh Alemu.

**Data curation:** Getaneh Alemu, Endalkachew Nibret, Abaineh Munshea, Melaku Anegagrie, Arancha Amor.

**Formal analysis:** Getaneh Alemu, Endalkachew Nibret, Abaineh Munshea.

**Investigation:** Getaneh Alemu, Arancha Amor.

**Methodology:** Getaneh Alemu, Endalkachew Nibret, Abaineh Munshea, Arancha Amor.

**Project administration:** Melaku Anegagrie.

**Resources:** Melaku Anegagrie.

**Software:** Getaneh Alemu, Endalkachew Nibret, Abaineh Munshea, Arancha Amor.

**Supervision:** Melaku Anegagrie, Arancha Amor.

**Validation:** Getaneh Alemu, Endalkachew Nibret, Arancha Amor.

**Visualization:** Abaineh Munshea, Melaku Anegagrie, Arancha Amor.

**Writing – original draft:** Getaneh Alemu.

**Writing – review & editing:** Endalkachew Nibret, Abaineh Munshea, Melaku Anegagrie, Arancha Amor.

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
