## [Decision Letter · Decision Letter 0]

4 Oct 2024

Dear Dr. Abebe,

Thank you for submitting your manuscript to PLOS ONE. After careful consideration, we feel that it has merit but does not fully meet PLOS ONE’s publication criteria as it currently stands. Therefore, we invite you to submit a revised version of the manuscript that addresses the points raised during the review process.

We look forward to receiving your revised manuscript.

Kind regards,

David Zadock Munisi, Ph.D

Academic Editor

PLOS ONE

2. We note that your Data Availability Statement is currently as follows: [All data are available in the manuscript]

3. We note that there is identifying data in the Supporting Information file <Quality assesment.docx>. Due to the inclusion of these potentially identifying data, we have removed this file from your file inventory. Prior to sharing human research participant data, authors should consult with an ethics committee to ensure data are shared in accordance with participant consent and all applicable local laws.

-Location data

Reviewers' comments:

Reviewer's Responses to Questions

**Comments to the Author**

1. Is the manuscript technically sound, and do the data support the conclusions?

Reviewer #1: Yes

Reviewer #2: Yes

Reviewer #3: Yes

Reviewer #4: Yes

Reviewer #5: Yes

Reviewer #6: Yes

2. Has the statistical analysis been performed appropriately and rigorously?

Reviewer #1: I Don't Know

Reviewer #2: No

Reviewer #3: I Don't Know

Reviewer #4: Yes

Reviewer #5: Yes

Reviewer #6: N/A

3. Have the authors made all data underlying the findings in their manuscript fully available?

Reviewer #1: Yes

Reviewer #2: No

Reviewer #3: Yes

Reviewer #4: Yes

Reviewer #5: Yes

Reviewer #6: Yes

4. Is the manuscript presented in an intelligible fashion and written in standard English?

Reviewer #1: Yes

Reviewer #2: Yes

Reviewer #3: Yes

Reviewer #4: Yes

Reviewer #5: Yes

Reviewer #6: Yes

Reviewer #1: Introduction

Please recast here ‘’The disease is endemic in 70 developing countries, and more than 200

million people are infected worldwide’’.

Please recast here and don’t start with ‘’Both’’ Both the intestinal (caused by S. mansoni) and

urogenital (caused by S. haematobium) forms of SCH are prevalent in Ethiopia [4].

Please recast here and break into two sentences ‘’Mass drug administration (MDA), facility-based case diagnosis and treatment, snail control in hotspots, SBCC, and access to safe water supply, basic sanitation, and hygiene (WASH) are the five strategies identified to achieve this goal [4].’’

Please recast here ‘’Even though, a few studies were conducted in Ethiopia, the previous review did not include them with a possible reason of robust eligibility criteria and review period’’

Discussion

Please recast here ‘’Apart from inadequate health education and promotion, socio-cultural factors might also be barriers to changes in attitude and practices towards SCH control and prevention [15].’’

Please recast here ‘’According to this review, comprehensive knowledge about sources of infection, routes and modes of transmission, morbidity, treatment, and prevention of SCH is lacking in Ethiopia.’’

Please recast here ‘’For instance, we observed that school and community-based education sessions

were conducted only at the time of deworming which is an inadequate method to establish good knowledge and change community behavior. ‘’

Please recast here ‘’If it is properly implemented, health education has a high input-output

ratio and is a cost-effective prevention and control method’’.

Please recast here ‘’Having comprehensive knowledge contributes to having a positive attitude, which in turn contributes to behavioral changes like reducing risky activities’’

Please recast here ‘’Variations in the health education implementation at different transmission settings and study populations might contribute to differences in behavioral changes in Ethiopia.’’

Conclusions

Please recast here ‘’Findings in the reviewed studies showed inadequate KAP level of people, despite both community and school-based health education platforms were established and have been implemented for more than a decade’’.

Reviewer #2: General Reviewer Observations and Comments

A systematic review on the knowledge, attitude, and practice (KAP) regarding schistosomiasis in Ethiopia reveals several critical insights. Generally, knowledge about schistosomiasis is variable, often influenced by factors such as education, geographic location, and access to health information.

Many communities have limited understanding of the transmission dynamics, symptoms, and prevention strategies associated with the disease. This gap in knowledge frequently translates to negative attitudes towards preventive measures, with some cultural beliefs affecting perceptions of the disease.

Moreover, behavioral practices associated with schistosomiasis prevention, including safe water usage and sanitation, are often inadequate.

Lack of access to clean water and proper sanitation facilities exacerbates the situation, contributing to the persistence of the disease in endemic regions.

Comments for review

• There is a lot of systematic editing and reviewing required from the investigator.

• When conducting research or a systematic review on knowledge, attitude, and practice (KAP) regarding schistosomiasis, various statistical analysis methods and data presentation tools should be used which are missing in the Data Analaysis section:

• Gaps in statistical analysis methods and the team could have used one of the following:

o Descriptive Statistics: Mean, Median, Mode: To summarize central tendencies in knowledge scores or attitudes.

o Frequency and Percentage: To detail respondents' knowledge levels, attitudes, and practices concerning schistosomiasis.

o Inferential Statistics: Chi-Square Test: To examine associations between categorical variables (e.g., knowledge levels across different demographic groups).

o T-Tests or ANOVA: To compare means of knowledge or attitude scores between different groups, such as educational levels or regions.

o Correlation Analysis: To assess relationships between knowledge and practice scores using Pearson or Spearman correlation coefficients.

o Regression Analysis: To identify factors influencing practices regarding schistosomiasis (e.g., logistic regression for binary outcomes).

o Factor Analysis: To identify underlying relationships between different KAP components and determine key factors affecting knowledge and practices.

Gaps in Data Presentation/Graphs/Charts and the team could have used one of the following:

o Tables: For presenting descriptive statistics, frequency distributions, and results of inferential analyses clearly.

o Bar Charts: To represent categories of knowledge and attitudes visually.

o Pie Charts: For displaying proportions of respondents' attitudes or practices.

o Box Plots: To show distributions of knowledge scores across different groups.

o Geospatial Analysis Tools: If location data is available, tools like ArcGIS can visualize schistosomiasis prevalence and KAP data across different geographic areas.

Using these statistical methods and presentation tables would have enhanced the clarity and impact of the findings related to KAP on schistosomiasis in Ethiopia.

Reviewer #3: Dear Author, thank you for submitting your interesting review. I think the data reported are of great interest and presented in an adequate fashion. Written English does not need corrections. Discussion and Introduction are appropriate.

Reviewer #4: I hereby recommend the manuscript to be published in the Plos One journal. The article on the systematic review of "Knowledge, attitude and practice on schistosomiasis in Ethiopia" has been extensively analyzed based on the studies on the disease between 2006 and 2023 in Ethiopia. Although the analysis revealed inadequate KAP on the disease but the publication of this research will further stress the need for the execution of health education and constant surveillance of schistosomiasis in Ethiopia for sustained prevention and control of the disease.

Reviewer #5: Important work

Complements the previous information.

Findings in the reviewed studies showed inadequate KAP level of people, despite both community and school-based health education platforms were established and have been implemented for more than a decade. This reminds for future strengthening the health education program and monitoring the practical implementation and impact assessment of the existing programs. Also, the implementation of two programs together (namely, STH and SCH), should be precise enough to avoid confusion and mixed information. Preparing a standard context-specific KAP assessment tool could be an immediate responsibility of the FMoH NTD prevention and control program. More studies are also recommended both at school and community level to explore the KAP level and associated factors at different endemicity settings among all

at-risk population groups.

Reviewer #6: The authors need to clarify a statement where four studies were referred but five results were presented under the section on knowledge about schistosomiasis from the papers reviewed. A comment on this was inserted. The language style was for the most part very passive. Although this is a review article, it is advised that some of the style be revised to be more active, a few of this has been addressed.

**Do you want your identity to be public for this peer review?** For information about this choice, including consent withdrawal, please see our Privacy Policy

Reviewer #1: **Yes: ** Prof Charles Oluwaseun Adetunji

Reviewer #2: No

Reviewer #3: No

Reviewer #4: **Yes: ** Aanuoluwa A. Adelani

Reviewer #5: **Yes: ** Ana Júlia Pinto Fonseca Sieuve Afonso

Reviewer #6: No

---

## [Author Response · Author response to Decision Letter 1]

30 Oct 2024

Responses to Reviewers’ Comments and Questions

Dear editor and reviewers, thank you for your constructive comments which all are important inputs for the betterment of the manuscript. Below we tried to respond to all the comments/questions one by one; we also have incorporated all the corrections in the revised manuscript (shown as highlighted).

Editor comments

Response: We have prepared and revised the manuscript based on the journal style requirements

2. We note that your Data Availability Statement is currently as follows: [All data are available in the manuscript]. Please confirm at this time whether or not your submission contains all raw data required to replicate the results of your study.

Response: we accept this comment. All extracted data are presented in Table 1 in the manuscript.

An excel data containing list of screened articles (after removing duplicates) and screening result (inclusion/exclusion) based on the title, abstract or full text, together with reasons for excluding studies, is submitted in supplementary file S1. In addition, a table containing name of data extractors, extraction date, and raw data extracted is submitted in Supplementary file S3. We have no other data available. Based on this, we have revised the Data availability statement in the revised submission.

3. We note that there is identifying data in the Supporting Information file <Quality assesment.docx>.

Response: To avoid identifying data, we have removed the author names and references, and we have coded included studies by letters (A to J).

4. Please review your reference list to ensure that it is complete and correct.

Response: We have reviewed the reference list to make it complete and correct. Fortunately, all our references are not retracted.

5. As required by our policy on Data Availability, please ensure your manuscript or supplementary information includes the following:

A numbered table of all studies identified in the literature search, including those that were excluded from the analyses. For every excluded study, the table should list the reason(s) for exclusion.

Response: we have submitted this in an excel file as supplementary file S1

Response: all included studies are published

Response: comment accepted and we have submitted the table containing all the requested contents as Supplementary file S3.

Response: No data or supporting information was obtained from other sources than the eligible reviewed studies.

If applicable for your analysis, a table showing the completed risk of bias and quality/certainty assessments for each study or outcome. Please ensure this is provided for each domain or parameter assessed. For example, if you used the Cochrane risk-of-bias tool for randomized trials, provide answers to each of the signaling questions for each study. If you used GRADE to assess certainty of evidence, provide judgements about each of the quality of evidence factor. This should be provided for each outcome.

Response: we have submitted a table containing quality assessment results of each reviewed study in Supplementary file S2.

Response: not applicable for our review, because we didn’t do quantitative analysis for any outcome.

Reviewers' comments to the author:

1. Is the manuscript technically sound, and do the data support the conclusions?

Reviewer #1: Yes

Reviewer #2: Yes

Reviewer #3: Yes

Reviewer #4: Yes

Reviewer #5: Yes

Reviewer #6: Yes

2. Has the statistical analysis been performed appropriately and rigorously?

Reviewer #1: I Don't Know

Reviewer #2: No

Reviewer #3: I Don't Know

Reviewer #4: Yes

Reviewer #5: Yes

Reviewer #6: N/A

Response: We have summarized data only based on thematic analysis. As we explained in the ‘Limitations’ section, due to differences in KAP assessment tools across reviewed studies, we couldn’t do meta-analysis or regression to present pooled results.

3. Have the authors made all data underlying the findings in their manuscript fully available?

Reviewer #1: Yes

Reviewer #2: No

Reviewer #3: Yes

Reviewer #4: Yes

Reviewer #5: Yes

Reviewer #6: Yes

Response: extracted data are presented in table 1 in the manuscript. Additional data (list of studies screened studies and their eligibility results, included studies quality assessment result, and extracted data) are submitted as supplementary file S1, S2 and S3.

4. Is the manuscript presented in an intelligible fashion and written in Standard English?

Reviewer #1: Yes

Reviewer #2: Yes

Reviewer #3: Yes

Reviewer #4: Yes

Reviewer #5: Yes

Reviewer #6: Yes

Response: More language editions are made based on comments attached by reviewers 2 and 6.

5. Review Comments to the Author

Reviewer #1:

Introduction

Please recast here ‘’The disease is endemic in 70 developing countries, and more than 200

million people are infected worldwide’’.

Response: comment accepted and the sentence is rephrased.

Please recast here and don’t start with ‘’Both’’ Both the intestinal (caused by S. mansoni) and

urogenital (caused by S. haematobium) forms of SCH are prevalent in Ethiopia [4].

Response: comment accepted and the sentence is rephrased.

Please recast here and break into two sentences ‘’Mass drug administration (MDA), facility-based case diagnosis and treatment, snail control in hotspots, SBCC, and access to safe water supply, basic sanitation, and hygiene (WASH) are the five strategies identified to achieve this goal [4].’’

Response: We have made language editions but despite we understand it is a long sentence, we prefer to make it one sentence as it contains a list of equally valued interventions.

Please recast here ‘’Even though, a few studies were conducted in Ethiopia, the previous review did not include them with a possible reason of robust eligibility criteria and review period’’

Response: comment accepted and the sentence is rephrased in the revised submission

Discussion

Please recast here ‘’Apart from inadequate health education and promotion, socio-cultural factors might also be barriers to changes in attitude and practices towards SCH control and prevention [15].’’

Response: comment accepted and the sentence is rephrased in the revised submission

Please recast here ‘’According to this review, comprehensive knowledge about sources of infection, routes and modes of transmission, morbidity, treatment, and prevention of SCH is lacking in Ethiopia.’’

Response: comment accepted and the sentence is rephrased in the revised submission

Please recast here ‘’For instance, we observed that school and community-based education sessions

were conducted only at the time of deworming which is an inadequate method to establish good knowledge and change community behavior. ‘’

Response: comment accepted and the sentence is rephrased in the revised submission

Please recast here ‘’If it is properly implemented, health education has a high input-output

ratio and is a cost-effective prevention and control method’’.

Response: comment accepted and the sentence is rephrased in the revised submission

Please recast here ‘’Having comprehensive knowledge contributes to having a positive attitude, which in turn contributes to behavioral changes like reducing risky activities’’

Response: comment accepted and the sentence is rephrased in the revised submission

Please recast here ‘’Variations in the health education implementation at different transmission settings and study populations might contribute to differences in behavioral changes in Ethiopia.’’

Response: comment accepted and the sentence is rephrased in the revised submission

Conclusions

Please recast here ‘’Findings in the reviewed studies showed inadequate KAP level of people, despite both community and school-based health education platforms were established and have been implemented for more than a decade’’.

Response: comment accepted and the sentence is rephrased in the revised submission

Reviewer #2: General Reviewer Observations and Comments

A systematic review on the knowledge, attitude, and practice (KAP) regarding schistosomiasis in Ethiopia reveals several critical insights. Generally, knowledge about schistosomiasis is variable, often influenced by factors such as education, geographic location, and access to health information.

Many communities have limited understanding of the transmission dynamics, symptoms, and prevention strategies associated with the disease. This gap in knowledge frequently translates to negative attitudes towards preventive measures, with some cultural beliefs affecting perceptions of the disease.

Moreover, behavioral practices associated with schistosomiasis prevention, including safe water usage and sanitation, are often inadequate.

Lack of access to clean water and proper sanitation facilities exacerbates the situation, contributing to the persistence of the disease in endemic regions.

Comments for review

• There is a lot of systematic editing and reviewing required from the investigator.

Response: comment accepted and we have made intense language editions based on suggestions in the attached PDF file from reviewer 2 and MS word doc attached by reviewer 6 (all editions are highlighted).

• When conducting research or a systematic review on knowledge, attitude, and practice (KAP) regarding schistosomiasis, various statistical analysis methods and data presentation tools should be used which are missing in the Data Analaysis section:

• Gaps in statistical analysis methods and the team could have used one of the following:

o Descriptive Statistics: Mean, Median, Mode: To summarize central tendencies in knowledge scores or attitudes.

o Frequency and Percentage: To detail respondents' knowledge levels, attitudes, and practices concerning schistosomiasis.

o Inferential Statistics: Chi-Square Test: To examine associations between categorical variables (e.g., knowledge levels across different demographic groups).

o T-Tests or ANOVA: To compare means of knowledge or attitude scores between different groups, such as educational levels or regions.

o Correlation Analysis: To assess relationships between knowledge and practice scores using Pearson or Spearman correlation coefficients.

o Regression Analysis: To identify factors influencing practices regarding schistosomiasis (e.g., logistic regression for binary outcomes).

o Factor Analysis: To identify underlying relationships between different KAP components and determine key factors affecting knowledge and practices.

Gaps in Data Presentation/Graphs/Charts and the team could have used one of the following:

o Tables: For presenting descriptive statistics, frequency distributions, and results of inferential analyses clearly.

o Bar Charts: To represent categories of knowledge and attitudes visually.

o Pie Charts: For displaying proportions of respondents' attitudes or practices.

o Box Plots: To show distributions of knowledge scores across different groups.

o Geospatial Analysis Tools: If location data is available, tools like ArcGIS can visualize schistosomiasis prevalence and KAP data across different geographic areas.

Using these statistical methods and presentation tables would have enhanced the clarity and impact of the findings related to KAP on schistosomiasis in Ethiopia.

Response: Dear reviewer, we appreciate your detail explanation and suggestion regarding statistical analysis and we have understood your concern very well. However, we didn’t use statistical analysis in this review because reviewed studies included varied study population (school-aged children, community, mothers of pre-school-aged children), and tools used to measure KAP varied across reviewed studies. Therefore, generating pooled results using meta-analysis or other statistical analyses is impossible. Rather a thematic analysis was used to present summarized results of reviewed studies. We have mentioned this in the ‘limitations’ section of the manuscript. Please check that similar reviews were published following the same method as ours (https://pubmed.ncbi.nlm.nih.gov/29347919/)

Reviewer #3: Dear Author, thank you for submitting your interesting review. I think the data reported are of great interest and presented in an adequate fashion. Written English does not need corrections. Discussion and Introduction are appropriate.

Response: dear reviewer, thank you for reviewing our manuscript, and for giving credit for our work

Reviewer #4: I hereby recommend the manuscript to be published in the Plos One journal. The article on the systematic review of "Knowledge, attitude and practice on schistosomiasis in Ethiopia" has been extensively analyzed based on the studies on the disease between 2006 and 2023 in Ethiopia. Although the analysis revealed inadequate KAP on the disease but the publication of this research will further stress the need for the execution of health education and constant surveillance of schistosomiasis in Ethiopia for sustained prevention and control of the disease.

Response: dear reviewer, thank you for reviewing our manuscript, and recommendation for publication in PLoS One.

Reviewer #5: Important work

Complements the previous information.

Findings in the reviewed studies showed inadequate KAP level of people, despite both community and school-based health education platforms were established and have been implemented for more than a decade. This reminds for future strengthening the health education program and monitoring the practical implementation and impact assessment of the existing programs. Also, the implementation of two programs together (namely, STH and SCH), should be precise enough to avoid confusion and mixed information. Preparing a standard context-specific KAP assessment tool could be an immediate responsibility of the FMoH NTD prevention and control program. More studies are also recommended both at school and community level to explore the KAP level and associated factors at different endemicity settings among all at-risk population groups.

Response: dear reviewer, thank you for reviewing our manuscript

Reviewer #6: The authors need to clarify a statement where four studies were referred but five results were presented under the section on knowledge about schistosomiasis from the papers reviewed. A comment on this was inserted. The language style was for the most part very passive. Although this is a review article, it is advised that some of the style be revised to be more active, a few of this has been addressed.

Response for comments from reviewer 6:

A. Dear reviewer, please accept our apology for the write up error under the section “knowledge about schistosomiasis”. The result ‘1.7% [25]’ was wrongly included and we have removed it in the revised manuscript. To avoid such other mistakes, we have double checked all the results from the extracted data table and from reviewed papers.

B. We have accepted and incorporated all grammar corrections given by track change in the attached manuscript from yo

---

## [Decision Letter · Decision Letter 1]

10 Jan 2025

Dear Dr. Abebe,

We look forward to receiving your revised manuscript.

Kind regards,

David Zadock Munisi, Ph.D

Academic Editor

PLOS ONE

Journal Requirements:

Reviewers' comments:

Reviewer's Responses to Questions

**Comments to the Author**

Reviewer #2: (No Response)

Reviewer #7: All comments have been addressed

Reviewer #8: (No Response)

2. Is the manuscript technically sound, and do the data support the conclusions?

Reviewer #2: Partly

Reviewer #7: Yes

Reviewer #8: Partly

3. Has the statistical analysis been performed appropriately and rigorously?

Reviewer #2: No

Reviewer #7: Yes

Reviewer #8: No

4. Have the authors made all data underlying the findings in their manuscript fully available?

Reviewer #2: No

Reviewer #7: Yes

Reviewer #8: Yes

5. Is the manuscript presented in an intelligible fashion and written in standard English?

Reviewer #2: Yes

Reviewer #7: Yes

Reviewer #8: Yes

Reviewer #2: Generally, knowledge about schistosomiasis is variable, often influenced by factors such as education, geographic location, and access to health information.

Many communities have a limited understanding of the transmission dynamics, symptoms, and prevention strategies associated with the disease. This gap in knowledge frequently translates to negative attitudes towards preventive measures, with some cultural beliefs affecting perceptions of the disease. Moreover, behavioral practices associated with schistosomiasis prevention, including safe water usage and sanitation, are often inadequate. Lack of access to clean water and proper sanitation facilities exacerbates the situation, contributing to the persistence of the disease in endemic regions.

Comments for review

There is a lot of systematic editing and reviewing required from the investigator. When conducting research or a systematic review on knowledge, attitude, and practice (KAP) regarding schistosomiasis, various statistical analysis methods and data presentation tools should be used which are missing in the Data Analysis section

Gaps in statistical analysis methods and the team could have used one of the following:

o Descriptive Statistics: Mean, Median, Mode: To summarize central tendencies in knowledge scores or attitudes.

oFrequency and Percentage: To detail respondents' knowledge levels, attitudes, and practices concerning schistosomiasis.

oInferential Statistics: Chi-Square Test: To examine associations between categorical variables (e.g., knowledge levels across different demographic groups).

oT-Tests or ANOVA: To compare means of knowledge or attitude scores between different groups, such as educational levels or regions.

oCorrelation Analysis: To assess relationships between knowledge and practice scores using Pearson or Spearman correlation coefficients.

oRegression Analysis: To identify factors influencing practices regarding schistosomiasis (e.g., logistic regression for binary outcomes).

oFactor Analysis: To identify underlying relationships between different KAP components and determine key factors affecting knowledge and practices.

Gaps in Data Presentation/Graphs/Charts and the team could have used one of the following:

oTables: For presenting descriptive statistics, frequency distributions, and results of inferential analyses clearly.

oBar Charts: To represent categories of knowledge and attitudes visually.

oPie Charts: For displaying proportions of respondents' attitudes or practices.

oBox Plots: To show distributions of knowledge scores across different groups.

oGeospatial Analysis Tools: If location data is available, tools like ArcGIS can visualize schistosomiasis prevalence and KAP data across different geographic areas.

Using these statistical methods and presentation tables would have enhanced the clarity and impact of the findings related to KAP on schistosomiasis in Ethiopia.

Reviewer #7: Dear Editor,

Thank you very much for sending me the revised article titled " Knowledge, attitude and practice on schistosomiasis in Ethiopia. A systematic review " for re-review.

I would like to inform you that in my opinion, the author has responded to all the comments in the revised article and this article is worthy of publication in a valuable journal.

Reviewer #8: (No Response)

**Do you want your identity to be public for this peer review?** For information about this choice, including consent withdrawal, please see our Privacy Policy

Reviewer #2: No

Reviewer #7: No

Reviewer #8: No

---

## [Author Response · Author response to Decision Letter 2]

13 Jan 2025

Responses to Reviewers’ Comments and Questions

Dear editor and reviewers, thank you for your action and comments based on our revised manuscript.

Below we tried to respond to all the comments/questions one by one; we also have incorporated all the corrections in the revised manuscript (shown as highlighted).

Comments to the Author

Reviewer #2:

Generally, knowledge about schistosomiasis is variable, often influenced by factors such as education, geographic location, and access to health information. Many communities have a limited understanding of the transmission dynamics, symptoms, and prevention strategies associated with the disease. This gap in knowledge frequently translates to negative attitudes towards preventive measures, with some cultural beliefs affecting perceptions of the disease. Moreover, behavioral practices associated with schistosomiasis prevention, including safe water usage and sanitation, are often inadequate. Lack of access to clean water and proper sanitation facilities exacerbates the situation, contributing to the persistence of the disease in endemic regions.

Response: dear reviewer, we totally accept your impression and we also believe knowledge, attitudes and preventive practices are closely inter-related.

Comments for review

There is a lot of systematic editing and reviewing required from the investigator. When conducting research or a systematic review on knowledge, attitude, and practice (KAP) regarding schistosomiasis, various statistical analysis methods and data presentation tools should be used which are missing in the Data Analysis section

Gaps in statistical analysis methods and the team could have used one of the following:

o Descriptive Statistics: Mean, Median, Mode: To summarize central tendencies in knowledge scores or attitudes.

o Frequency and Percentage: To detail respondents' knowledge levels, attitudes, and practices concerning schistosomiasis.

o Inferential Statistics: Chi-Square Test: To examine associations between categorical variables (e.g., knowledge levels across different demographic groups).

o T-Tests or ANOVA: To compare means of knowledge or attitude scores between different groups, such as educational levels or regions.

oCorrelation Analysis: To assess relationships between knowledge and practice scores using Pearson or Spearman correlation coefficients.

oRegression Analysis: To identify factors influencing practices regarding schistosomiasis (e.g., logistic regression for binary outcomes).

oFactor Analysis: To identify underlying relationships between different KAP components and determine key factors affecting knowledge and practices.

Gaps in Data Presentation/Graphs/Charts and the team could have used one of the following:

oTables: For presenting descriptive statistics, frequency distributions, and results of inferential analyses clearly.

oBar Charts: To represent categories of knowledge and attitudes visually.

oPie Charts: For displaying proportions of respondents' attitudes or practices.

oBox Plots: To show distributions of knowledge scores across different groups.

oGeospatial Analysis Tools: If location data is available, tools like ArcGIS can visualize schistosomiasis prevalence and KAP data across different geographic areas.

Using these statistical methods and presentation tables would have enhanced the clarity and impact of the findings related to KAP on schistosomiasis in Ethiopia.

Response: Dear reviewer, based on other reviewers’ and our own later evaluation of the review, we agreed that the review we conducted should be “ a scoping review” rather than “ a systematic review” and we have made this correction to the revised manuscript. Because, reviewed studies included varied study population (school-aged children, community, mothers of pre-school-aged children), and tools used to measure KAP varied across reviewed studies; i.e. not focused to a specific question targeting a certain study group, and tools used in included studies varied. Therefore, generating pooled results using meta-analysis or other statistical analyses, which are requirements of a systematic review. Rather a thematic analysis was used to present summarized results of reviewed studies. We have mentioned this in the ‘limitations’ section of the manuscript. Thematic analysis is a feature of thematic or scoping review.

Reviewer #7: Dear Editor,

Thank you very much for sending me the revised article titled " Knowledge, attitude and practice on schistosomiasis in Ethiopia. A systematic review " for re-review.

I would like to inform you that in my opinion, the author has responded to all the comments in the revised article and this article is worthy of publication in a valuable journal.

Response: Thank you for your constructive comments

Reviewer #8: (No Response)

Attachment comment

Plos one Review

Title: Knowledge, attitude and practice on schistosomiasis in Ethiopia. A systematic review

I am not convinced the selected studies were illegible for a systematic review.

Is this a scoping review or a systematic review?

-The authors lack knowledge of the difference between scoping review and systematic review

May I request that the authors decide if they want to do a scoping or systematic review using the following table?

Feature Systematic Review Scoping Review

Purpose Answer a specific, focused research question Map the breadth and scope of research

Research Question Specific, focused, often hypothesis-driven Broad, exploratory

Methodology Structured, with critical appraisal and synthesis Flexible, with categorisation and mapping

Data Analysis Quantitative synthesis (e.g., meta-analysis) Qualitative summary, no pooling of data

Outcome Evidence-based conclusion or recommendation Overview of research landscape, gaps, and trends

Usefulness Decision-making for practice or policy Identifying research gaps, informing future reviews

Response: Thank you for the comment with detailed explanation. Based on your explanation and discussion among authors, we authors agreed that our review is “a scoping review” rather than “a systematic review”. We have made this correction to the revised submission. Despite we followed PRISMA guideline to search and review literatures, the data we extracted couldn’t be analysed in a way required for systematic reviews. Because, reviewed studies varied by study population (school-aged children, community, mothers of pre-school-aged children), and tools used to measure KAP varied across reviewed studies; i.e. not focused to a specific question targeting a certain study group, and tools used in included studies varied.

Abstract

Background: the first sentence is not specific to the main focus of the review, which is Schistosomiasis

Response: we accept this comment and we have revised the sentence (highlighted)

Methods:

- which framework was used to guide the systematic review?

- What search strategy was used to search literature in electronic databases

- Which databases were selected and why?

- What was the duration of the search

- What was used to document the selection process

Response: We have included all the above queries in the ‘methods’ section of the abstract in the revised manuscript. But we didn’t justify why we use the mentioned data bases just to avoid lengthy abstract. For your information, we used those mentioned electronic search databases because those are the only data bases we accessed freely.

Results: the results of the review are presented quantitatively

- It is also a requirement that authors mention the type of studies they found through the search

Response: all were cross-sectional studies (mentioned in the revised manuscript). This is also presented in ‘the type of study’ column in table 1.

What are the strengths and limitations of this systematic review?

Response: presented in the ‘strengths and limitations of the review’ section.

Methodology: Major revision is required

- A systematic review is a comprehensive, structured approach to synthesizing research evidence on a specific topic or question. It follows a clear, predefined methodology to minimize bias, ensuring the review is transparent, reproducible, and reliable. The current methodology is not articulated clearly. The following items are not fully described to a level where the manuscript can be accepted and published by a journal,

• There is no specific, clear, answerable research question. This typically follows the PICO framework (Population, Intervention, Comparison, Outcome) for clinical questions or other relevant frameworks for different fields.

According to the Authors, this review aimed to evaluate the KAP concerning SCH among different population groups in Ethiopia; this aim is written in the background. Another aim written in the methodology reads as follows: We aimed to assess KAP-level progress towards SCH in Ethiopia after the launch of the Health Extension Program

(2004–2005) what is the exact aim/ research question of this review

Response: The aim of the review was to evaluate the KAP concerning SCH among different population groups in Ethiopia. In the methodology, the phrase “ to assess KAP-level progress towards SCH in Ethiopia after the launch of the Health Extension Program” was put to explain why we include studies published only after 2005 (start of the health extension program); we have rephrased this to make it clear.

Registration: Many systematic reviews are registered with platforms like PROSPERO (for health-related reviews) to prevent duplication and ensure transparency; it's unclear if this review has been registered to avoid duplication

Response: we didn’t register the review in any such platform

Comprehensive Literature Search

• Database Selection: Reviews require Searching multiple relevant databases (e.g., PubMed, Scopus, Web of Science, Cochrane Library) to find studies related to the research question. Please revise accordingly.

• Search Strategy: The authors used Boolean only. What about medical subject headings (MeSH)?

• Gray Literature: the authors should minimise publication bias and consider including non-peer-reviewed sources like theses, conference proceedings, and government reports.

Response: Regarding databases, we didn’t search in scopus, web of science and cocrane because we have no access to those databases. We have made gray literatures but we didn’t get any unpublished eligible literatures.

Screen and Select Studies

Title and Abstract Screening: what were the inclusion and exclusion criteria for the review?

Full-text Screening: what were the eligibility criteria for the review?

Quality assessment versus Assessment of Risk of Bias

Normally, systematic reviews use an assessment of Risk of Bias: Evaluate the quality of studies using Cochrane's Risk of Bias tool to assess methodological rigour. I haven't seen this assessment.

It is common to have two independent reviewers screen studies to reduce bias. Were there any disagreements which required consensus or a third reviewer?

Assessment of the Study Quality

• Risk of Bias: how did the authors Evaluate the quality and risk of bias in the included studies, and which standardized tools did they use (e.g., Cochrane Risk of Bias tool, ROBINS-I for non-randomized studies)?

• Critical Appraisal: how did the authors appraise the included studies? Consider factors like study design, sample size, methodology, and funding sources that may influence the validity of the findings.

Synthesize the Data: Synthesis is insufficient due to the limited number of studies that have included data.

• Quantitative Synthesis (Meta-analysis): where the studies are not sufficiently homogeneous (in terms of design, participants, interventions, and outcomes), conduct a meta-analysis to combine the results quantitatively.

o Calculate effect sizes (e.g., risk ratios, mean differences) and assess study heterogeneity. The authors haven’t done this.

• Qualitative Synthesis: If the studies are too heterogeneous for a meta-analysis, provide a narrative synthesis, summarizing key findings across studies, highlighting patterns, and identifying gaps in evidence.

• Sensitivity Analyses: Conduct sensitivity analyses to assess how robust the findings are to changes in the analysis, such as excluding studies with a high risk of bias.

Assess the Quality of Evidence (GRADE)

As per the previous comments from the reviewers, the authors haven’t applied this.

Response: Overall, the nature of the literature we reviewed are not fully eligible to perform all the comments you provided here. That is why we revised the review to be a scoping review. So, we couldn’t address all your comments regarding data quality and analysis

---

## [Decision Letter · Decision Letter 2]

3 Feb 2025

Dear Dr. Abebe,

Thank you for submitting your manuscript to PLOS ONE. After careful consideration, we feel that it has merit but does not fully meet PLOS ONE’s publication criteria as it currently stands. Therefore, we invite you to submit a revised version of the manuscript that addresses the points raised during the review process.

We look forward to receiving your revised manuscript.

Kind regards,

David Zadock Munisi, Ph.D

Academic Editor

PLOS ONE

**Journal Requirements:**

Reviewers' comments:

Reviewer's Responses to Questions

**Comments to the Author**

Reviewer #2: All comments have been addressed

Reviewer #8: (No Response)

2. Is the manuscript technically sound, and do the data support the conclusions?

Reviewer #2: Yes

Reviewer #8: No

3. Has the statistical analysis been performed appropriately and rigorously?

Reviewer #2: Yes

Reviewer #8: N/A

4. Have the authors made all data underlying the findings in their manuscript fully available?

Reviewer #2: Yes

Reviewer #8: Yes

5. Is the manuscript presented in an intelligible fashion and written in standard English?

Reviewer #2: Yes

Reviewer #8: Yes

**Reviewer #2:**  The authors have provided point by point responses to the reviewer queries raised. This is can be moved to the next level for considerations.

**Reviewer #8: ** Title: Knowledge, attitude and practice on schistosomiasis in Ethiopia. A scoping review

- Authors must use a semicolon (:) to separate Ethiopia and A scoping review

Abstract

-Methodology: the authors didn’t use the framework for scoping reviews , this is one of the guidelines for writing a review as a results the are missing steps .

- there is a typo error in the last sentence of the methodology

Introduction

- The purpose of the scoping reviews is to map existing evidence, not to EVALUATE

- The authors mention that they conducted a systematic review; see the sentence below taken directly in the introduction.

“Therefore, the aim of this review was to evaluate the KAP in

relation to SCH among different population groups in Ethiopia. This systematic review will help

stakeholders assess the impact of the ongoing health education programs.”

Methodology:

The authors haven't used a framework for scoping reviews; as a result, there are missing steps.

**Do you want your identity to be public for this peer review?** For information about this choice, including consent withdrawal, please see our Privacy Policy

Reviewer #2: No

Reviewer #8: No

---

## [Author Response · Author response to Decision Letter 3]

7 Feb 2025

We have removed figure 1 from the manuscript

---

## [Decision Letter · Decision Letter 3]

7 Apr 2025

Dear Dr. Abebe,

We look forward to receiving your revised manuscript.

Kind regards,

David Zadock Munisi, Ph.D

Academic Editor

PLOS ONE

Journal Requirements:

Reviewers' comments:

Reviewer's Responses to Questions

**Comments to the Author**

Reviewer #7: All comments have been addressed

Reviewer #9: (No Response)

2. Is the manuscript technically sound, and do the data support the conclusions?

Reviewer #7: Yes

Reviewer #9: Partly

3. Has the statistical analysis been performed appropriately and rigorously?

Reviewer #7: I Don't Know

Reviewer #9: N/A

4. Have the authors made all data underlying the findings in their manuscript fully available?

Reviewer #7: Yes

Reviewer #9: No

5. Is the manuscript presented in an intelligible fashion and written in standard English?

Reviewer #7: Yes

Reviewer #9: No

Reviewer #7: Accept

all required questions have been answered and that all responses meet formatting specifications.

Reviewer #9: Generally, your document has merit but needs major revision. your need to address all the issues raised in your main document.

are you interested in KAP about the disease of its prevention and control practices? so, you need to clarify your title. it is on prevention, about SCH, etc.??? you need to clearly specify it.

the document has many typo errors and needs major edition.

**Do you want your identity to be public for this peer review?** For information about this choice, including consent withdrawal, please see our Privacy Policy

Reviewer #7: **Yes: ** Abdolreza Sotoodeh Jahromi

Reviewer #9: No

---

## [Author Response · Author response to Decision Letter 4]

7 Apr 2025

Dear editor and reviewers, thank you for your action and comments based on our revised manuscript.

Below we tried to respond to all the comments/questions one by one; we also have incorporated all the corrections in the revised manuscript (shown as highlighted).

Reviewer #7: Accept

all required questions have been answered and that all responses meet formatting specifications.

Reflection: Thank you for your positive response

Reviewer #9: Generally, your document has merit but needs major revision. your need to address all the issues raised in your main document. are you interested in KAP about the disease of its prevention and control practices? so, you need to clarify your title. it is on prevention, about SCH, etc.??? you need to clearly specify it. the document has many typo errors and needs major edition.

Response: We accept the comment and have made revision on the title. We were interested in knowledge and attitudes towards SCH, as well as, risky activities practiced by the community (not on prevention). We have made tense and grammar corrections throughout the document (all are highlighted).

---

## [Decision Letter · Decision Letter 4]

30 Sep 2025

Dear Dr. Abebe,

Thank you for submitting your manuscript to PLOS ONE. After careful consideration, we feel that it has merit but does not fully meet PLOS ONE’s publication criteria as it currently stands. Therefore, we invite you to submit a revised version of the manuscript that addresses the points raised during the review process.

We look forward to receiving your revised manuscript.

Kind regards,

Hammed Oladeji Mogaji, Ph.D

Academic Editor

PLOS ONE

Journal Requirements:

Additional Editor Comments:

Please revise the manuscript following the minor comments from the reviewers. While returning your revised manuscript, ensure you upload both the clean and tracked copy that has your changes.

Reviewers' comments:

Reviewer's Responses to Questions

**Comments to the Author**

Reviewer #7: All comments have been addressed

Reviewer #9: (No Response)

Reviewer #10: (No Response)

2. Is the manuscript technically sound, and do the data support the conclusions?

Reviewer #7: Yes

Reviewer #9: Partly

Reviewer #10: Partly

3. Has the statistical analysis been performed appropriately and rigorously?

Reviewer #7: I Don't Know

Reviewer #9: N/A

Reviewer #10: Yes

4. Have the authors made all data underlying the findings in their manuscript fully available?

Reviewer #7: Yes

Reviewer #9: Yes

Reviewer #10: Yes

5. Is the manuscript presented in an intelligible fashion and written in standard English?

Reviewer #7: Yes

Reviewer #9: No

Reviewer #10: Yes

Reviewer #7: Dear editor

Many thanks for your kind review invitation a manuscript " Knowledge, attitude and risky practice on schistosomiasis in Ethiopia: A scoping review ".

I would like to inform you that I had accepted the manuscript in revision-3 style.

Reviewer #9: Overall, the manuscript has a merit. to make the manuscript sound for readers, it is better if you consider my comments that I forwarded in the main manuscript.

Reviewer #10: 1. Title and Abstract

The phrase “risky practice” is grammatically incorrect; it should be “risky practices”.

The abstract is clear in stating the background and purpose. However, quantitative data should be provided where possible (e.g., number of studies included, participant characteristics).

Avoid vague statements like “studies found large differences…”—instead, provide specific findings or percentages for clarity.

2. Introduction

Clearly define what is meant by “risky practices” early on in this section. Is this referring to water contact, open defecation, etc.?

3. Methods

The search strategy is appropriate but needs more transparency. Provide the full search string, date last searched, and whether the search was peer-reviewed (PRESS standard).

Clarify why Google Scholar was included given its lower specificity for peer-reviewed literature.

4. Results

How were “good” or “poor” knowledge/attitudes/practices defined across studies? This lack of standardization complicates synthesis.

**Do you want your identity to be public for this peer review?** For information about this choice, including consent withdrawal, please see our Privacy Policy

Reviewer #7: No

Reviewer #9: No

Reviewer #10: No

---

## [Author Response · Author response to Decision Letter 5]

4 Oct 2025

Responses to Reviewers’ Comments and Questions

Dear editor and reviewers, thank you for your action and comments based on our revised manuscript.

Below we tried to respond to all the comments/questions one by one; we also have incorporated all the corrections in the revised manuscript (shown as highlighted).

Reviewer #7:

Dear editor, many thanks for your kind review invitation a manuscript " Knowledge, attitude and risky practice on schistosomiasis in Ethiopia: A scoping review ". I would like to inform you that I had accepted the manuscript in revision-3 style.

Response: thank you for accepting the manuscript

Reviewer #9:

Overall, the manuscript has a merit. To make the manuscript sound for readers, it is better if you consider my comments that I forwarded in the main manuscript.

Response: We accepted all the comments given in the main manuscript and have made revisions accordingly (all changes are highlighted in the revised submission).

Comments not accepted in the abstract

1. Do not use abbreviations in the abstract

Reason: It is not recommended to use abbreviations in the abstract; there is no hard rule that totally prohibit use of abbreviations in the abstract. Hence, we use a few abbreviations just to avoid lengthy abstract.

2. Try to include the discriptive statistcis used to report the result.

Reason: we just synthesized and presented the summarized data. We didn’t do any statistical analysis.

3. It is better if you report what you found. This is not summary. You should summarize the result you found. like 29% had poor attitude etc.

Reason: Reviewed studies used quite varied KAP assessment tools and the reporting was not consistent. So, it is difficult to pool results. Rather we presented main findings of each study separately.

In the study selection

Question: why did you exclude articles before 2006?

Response: Health education and promotion on SCH started in 2004 when the health extension program became operational in Ethiopia. So, we aim to assess the progress in KAP after the start of the health education program.

Questions in the results section

Previously. I have asked you to show whether this articles were retreived or not. do you think all this are removed because of duplicates?

Response: these articles were retrieved and all these were removed because they were duplicates. We used the less specific ‘ google scholar’ which makes large number of duplicates.

Reviewer #10:

1. Title and Abstract

The phrase “risky practice” is grammatically incorrect; it should be “risky practices”.

Response: we accept the comment revised accordingly

The abstract is clear in stating the background and purpose. However, quantitative data should be provided where possible (e.g., number of studies included, participant characteristics).

Response: We accept the comment and revised accordingly

Avoid vague statements like “studies found large differences…”—instead, provide specific findings or percentages for clarity.

Response: We accept the comment but because each study used different tools and inconsistent reporting, it is difficult to put pooled quantitative results.

2. Introduction

Clearly define what is meant by “risky practices” early on in this section. Is this referring to water contact, open defecation, etc.?

Response: yes, it is referring to practices that predispose for Schistosoma infection (water contact, open defication…)

3. Methods

The search strategy is appropriate but needs more transparency. Provide the full search string, date last searched, and whether the search was peer-reviewed (PRESS standard).

Response: We accept the comment and revised accordingly

Clarify why Google Scholar was included given its lower specificity for peer-reviewed literature.

Response: we agree that google scholar is less specific but we used it for literature search jus not to miss studies to be included.

4. Results

How were “good” or “poor” knowledge/attitudes/practices defined across studies? This lack of standardization complicates synthesis.

Response: Yes, there was inconsistency in defining good or poor KAP. We have explained this limitation (lack of standard cut off values) in the last paragraph of the ‘discussion’ section.

---

## [Editor Report · Decision Letter 5]

5 Nov 2025

Knowledge, attitude and risky practice on schistosomiasis in Ethiopia: A scoping review

PONE-D-24-22605R5

Dear Dr. Abebe,

We’re pleased to inform you that your manuscript has been judged scientifically suitable for publication and will be formally accepted for publication once it meets all outstanding technical requirements.

Kind regards,

Hammed Oladeji Mogaji, Ph.D

Academic Editor

PLOS ONE
---

## [Editor Report · Acceptance letter]

PONE-D-24-22605R5

PLOS ONE

Dear Dr. Abebe,

I'm pleased to inform you that your manuscript has been deemed suitable for publication in PLOS ONE. Congratulations! Your manuscript is now being handed over to our production team.

Kind regards,

on behalf of

Dr. Hammed Oladeji Mogaji

Academic Editor

PLOS ONE